

# Linkages between atmospheric rivers and humid heat across the United States

Colin Raymond[1], Anamika Shreevastava[2], Emily Slinskey[3], and Duane Waliser[2]

[1]Joint Institute for Regional Earth System Science and Engineering, University of California, Los Angeles, Los Angeles, 90095, USA
[2]Jet Propulsion Laboratory/California Institute of Technology, Pasadena, 91109, USA
[3]Department of Atmospheric and Oceanic Sciences, University of California, Los Angeles, Los Angeles, 90095, USA

*Correspondence to*: Colin Raymond (csraymond@ucla.edu)

**Abstract.** The global increase in atmospheric water vapour due to climate change tends to heighten the dangers associated with both humid heat and heavy precipitation. Process-linked correlations between these two extremes, particularly those which cause them to occur close together in space or time, are of special concern for efforts to understand and mitigate their impacts. Here we investigate how atmospheric rivers relate to the risk of summertime humid heat in the US. We find that the hazards of atmospheric rivers and humid heat often occur in close proximity, most notably across the northern third of the US. In this region, high levels of water vapour — resulting from the spatially organised horizontal moisture plumes that characterise atmospheric rivers — act to amplify humid heat, generally during the transition from dry high-pressure ridge conditions to wet low-pressure trough conditions. In contrast, the Southeast, Southwest, and Northwest US tend to experience atmospheric rivers and humid heat separately, representing an important negative correlation of joint risk.

## 1 Introduction

Hot and humid weather — prime conditions for heat stress — is increasing in occurrence and severity over most of the globe, a consequence of both rising temperature and specific humidity (Raymond et al. 2020; Buzan & Huber 2020). Several recent studies have found that wet and hot conditions can occur in rapid sequence, posing the compound threat of infrastructure damage followed by a public-health crisis to which response capacities are diminished (Zhang & Villarini 2020; Liao et al. 2021; Gu et al. 2022; Sauter et al. 2023), and more generally the challenge of enhanced impacts due to resource limitations from two damaging events happening close together in space or time (de Ruiter et al. 2020).

The joint wet-hot risk is underlain by physical connections in the form of both atmospheric-circulation patterns and land-surface feedbacks. Soil moisture is a particularly important modulator, with high-humid-heat days being favoured after wet days in arid areas of the subtropics (Liu et al. 2019; Speizer et al. 2022). Conversely, high temperatures are followed by an increased likelihood of precipitation in situations where there is a mechanism that facilitates or forces ascent, whether large-scale as in North China, Central Europe, or the Midwest US (Deng et al. 2020; You & Wang 2021; Sauter et al. 2023; Zhang & Villarini 2020) or mesoscale as in Florida, USA (Raghavendra et al. 2019). In the former cases, moisture





convergence occurs due to the same circulation regime that favours subsidence, anomalous radiation, and southerly flow. Similarly, the occurrence of successive heat and flood events on the Australian east coast has been attributed to a slight geographic shift in position of a ridge east of Queensland, with warm and humid onshore flow rapidly transitioning to hot and dry offshore flow that raises temperatures while moisture levels are already high (Sauter et al. 2022; Boschat et al.

2015). However, of the aforementioned studies, only Zhang and Villarini (2020) investigate humid heat, rather than high temperatures alone. Consideration of mechanisms for tripartite heat-vapour-precipitation connections has also been underdeveloped.

Atmospheric rivers [ARs] are broadly defined as long-distance conveyors of water vapour, serving to effect poleward moisture transport and also favouring high winds and heavy precipitation (Ralph et al. 2020; Ralph et al. 2018;

Guan & Waliser 2015; Neiman et al. 2008). Related terms from the literature which more precisely locate and describe vapour-transport features include warm conveyor belts (Madonna et al. 2014) and moist low-level jets (Ralph et al. 2018; Stensrud 1996). ARs have been almost exclusively discussed phenomenologically (Gimeno et al. 2021), and consequently a wide diversity of meteorological patterns are entrained into AR classification schemes, even within the same region and season. ARs are most closely related to maxima of moisture transport, otherwise known as integrated vapour transport [IVT],

which occur principally in connection with extratropical cyclones but also with deep monsoon-related flow and continental low-level jets, among other systems (de Vries 2021; Gimeno et al. 2021; Corringham et al. 2019). Notable instances of the latter two phenomena are located in the Midwest US, northern India, and southern South America (de Vries 2021). Although the first-described and best-known AR types occur in the extratropical cold season, warm-season varieties can have a substantial imprint on regional hydroclimate (Slinskey et al. 2020). Several recent papers have focused on the characteristics

of warm-season ARs in various parts of the world (Park et al. 2021; Guan & Waliser 2019; Nayak & Villarini 2017).

Recognizing this state of existing literature as well as the weather-system perspective that ARs offer with respect to ensuring the physical meaningfulness of risk relationships, we investigate here the spatiotemporal patterns of humid heat and ARs across the contiguous US, and in doing so explore the potential for ARs to encapsulate a strong and process-based link between humid heat, precipitation, and moisture transport.

## 2 Data and methods

### 2.1 Time period and regions

Our analysis spans 1980-2020, for the extended warm season of May-September, and relies primarily on variables from the 6-hourly MERRA-2 reanalysis (Gelaro et al. 2017) as described further below. We consider both the gridcell level and spatial means across seven regions of the contiguous US defined by the US National Climate Assessment: Northwest

[NW], Southwest [SW], Northern Great Plains [NGP], Southern Great Plains [SGP], Midwest [MW], Southeast [SE], and Northeast [NE] (Jay et al. 2018). These regions are included in Fig. 1.



## 2.2 Atmospheric rivers

For ARs, we use the MERRA-2-based Guan-Waliser AR-detection algorithm (Guan & Waliser 2019). This algorithm incorporates a percentile-based thresholding of IVT, as well as geometric and direction-of-motion criteria, to define AR presence/absence at each gridcell and timestep. We subsequently define AR gridcell-days as those for which an AR is present at a gridcell for at least two of that day's four timesteps. Each AR is also assigned a single intensity category for each day based on a scale of 1 (weak) to 5 (strong) (Ralph et al. 2019); we consider strong ARs to correspond to categories 4 and 5.


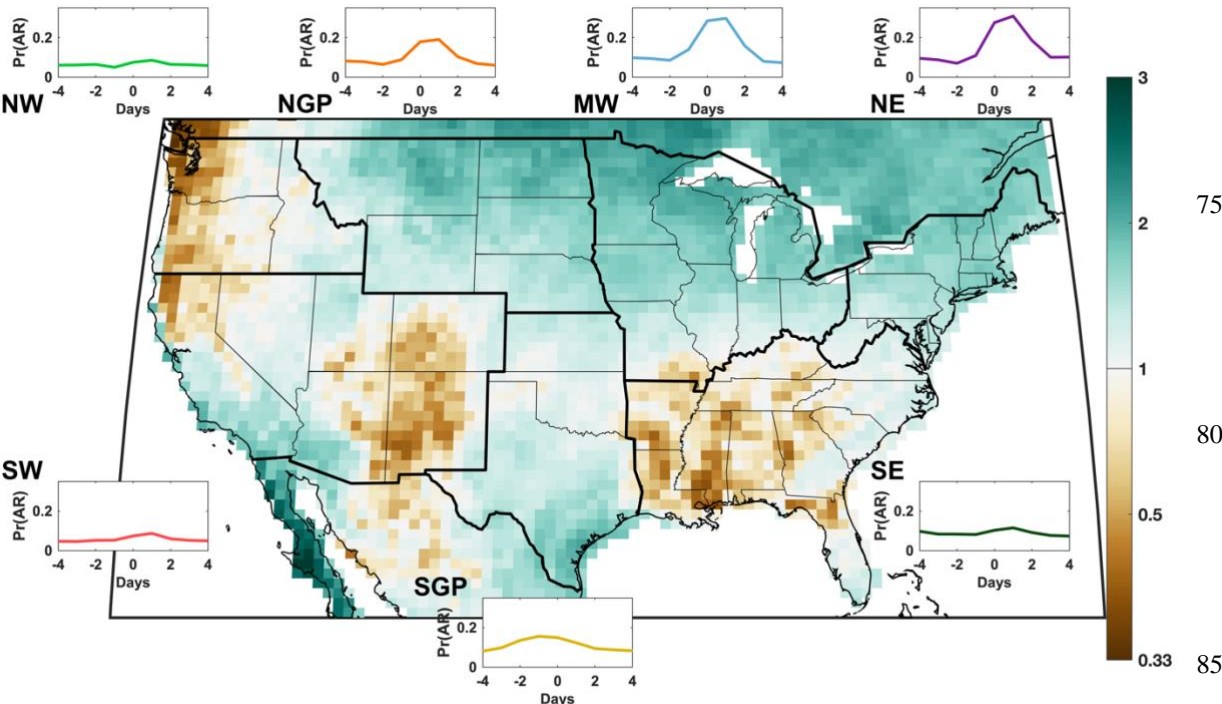

*Figure 1: AR/humid-heat interaction statistics*

*(Map) Relative risk of an AR occurring in close proximity (within 1 day and 100 km) to a humid-heat day at each gridcell. (Inset plots) For each region (black outlines), composited AR probability for the 9 days surrounding peak humid-heat days.*


## 2.3 Humid heat

To characterise humid heat, we use daily maxima of 2-m wet-bulb temperature [Tw], calculated from hourly dry-bulb temperature and dewpoint temperature (Davies-Jones 2008). We compute percentiles against a 30-day-smoothed
climatology at each gridcell, then define a 'humid-heat day' as a day with Tw above the 95th percentile. A 'peak humid-heat day' is a humid-heat day that additionally satisfies the constraints of having the highest Tw value within 3 days on either




side, as well as Tw having been below the 90th percentile within the preceding 3 days. This 'peak' framing is intended to capture sequences associated with high humid heat that has recently and notably intensified, as we wish to examine most closely the processes that exacerbate humid heat rather than those that prolong it. Lastly, 'regional humid-heat days' and 'regional peak humid-heat days' are fully analogous to their individual-gridcell equivalents but with each criterion applied instead to the mean of all gridcells in a region. We find that 1.6% of all May-September days are peak humid-heat days, or approximately 2.5 days per year on average; regional peak humid-heat days range in frequency from 1.1 per year (Southwest) to 2.8 per year (Northeast). Composites are then constructed as the mean across all days in a particular category.

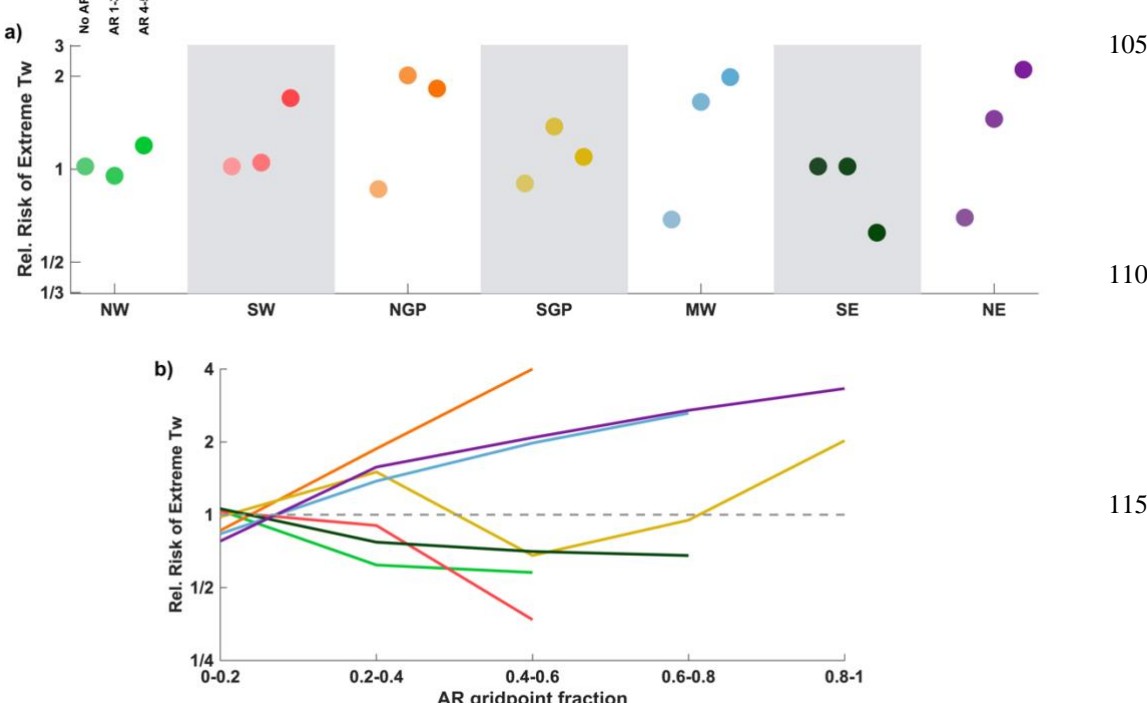

**Figure 2: Relative risk of AR/humid-heat interaction by AR intensity and extent**

*a) For each region, the relative risk of a humid-heat day with no AR nearby, i.e. within 1 day and 100 km (yellow); with an AR of category 1-3 nearby (blue); and with an AR of category 4-5 nearby (purple). b) Relative risk of humid heat between regional humid-heat days and an identical-Z500 set of days without regional humid heat (see Methods), binned by regional AR extent. Note that most regions lack any days with >80% regional AR coverage.*

### 2.4 Interaction between atmospheric rivers and humid heat

We define as 'interaction' between ARs and humid heat those cases where humid-heat days at a gridcell occur within 1 day and 100 km of an AR. This is also described in the text as an AR occurring in 'close proximity' to humid heat or 'nearby'. According to these definitions, 2.4% of all MJJAS gridcell-days across the US exhibit AR/humid-heat interaction.



Relative risk refers to the computed probability of such interaction on particular sets of days versus the probability expected if ARs and humid heat were randomly distributed relative to one another. As an additional metric, we assess regional AR/humid-heat interaction when controlling for anomalies in 500-hPa geopotential height [Z500] by comparing two sets of days: one comprising all regional-humid-heat days, the other comprising a random selection of warm-season days with

135    identical regional-mean Z500 anomalies but that do not meet the regional-humid-heat threshold.

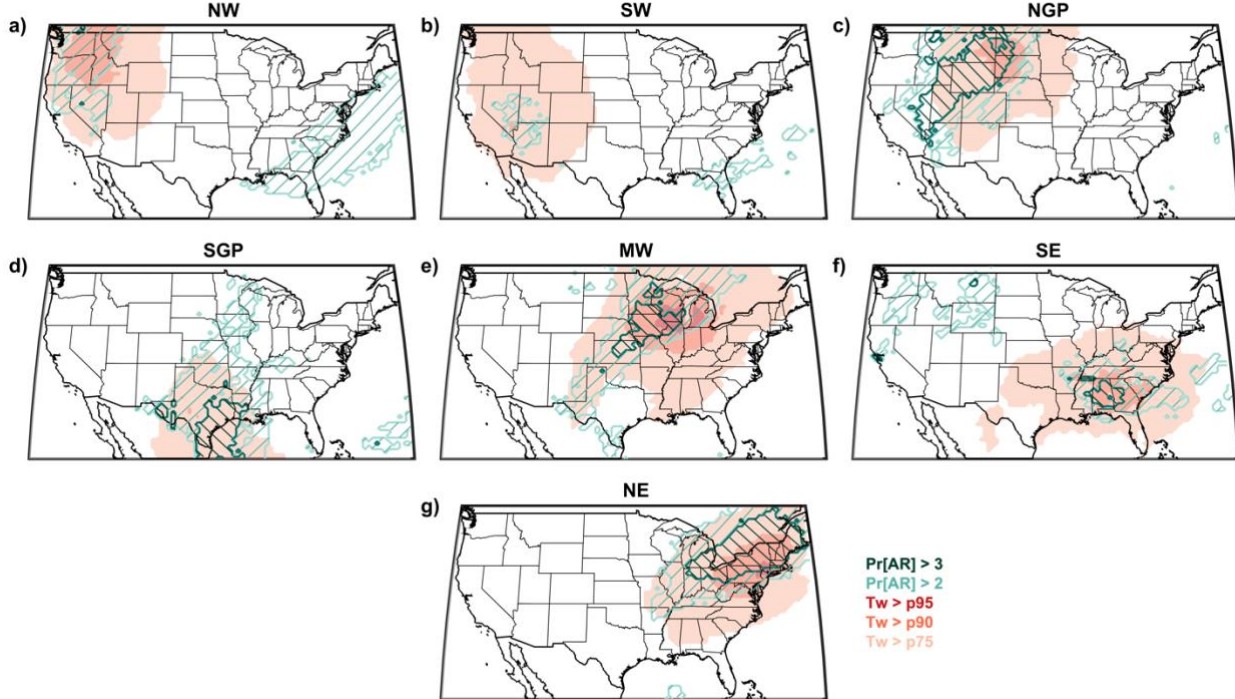

145    *Figure 3: AR/humid-heat interaction maps*

*AR/humid-heat interactions for each region: (a) Northwest, (b) Southwest, (c) Northern Great Plains, (d) Southern Great Plains, (e) Midwest, (f) Southeast, (g) Northeast. Shading shows where mean humid heat, for the composited humid-heat days, exceeds the MJJAS 95th percentile (dark red), 90th percentile (red), or 75th percentile (light red). Contours indicate where the AR relative likelihood within 2 days of these composited events exceeds 3 (dark teal) or 2 (light teal). Gridcells*

150    *with mean AR likelihood <10% are masked for reliability.*

# 3 Results

## 3.1 Region-specific AR/humid-heat interaction statistics

We find three primary areas where ARs and humid heat tend to interact: the northern tier of the US from Montana eastward; southeastern Texas; and the low elevations between central California and Arizona (Figure 1). In each area,



conditioned on humid-heat days, the probability of a nearby AR is at least doubled relative to chance. For brevity, in this study we henceforth consider only the first area, which is the largest and bears the most relevance to existing literature. Other parts of the country such as the Southeast, coastal Northwest, and high-mountain Southwest show a notable reduction in joint risk, with AR/humid-heat interactions being approximately half as likely as they would if the two hazards were unrelated. Where they occur, these interactions follow a clear temporal signature: relative to peak humid heat, ARs are typically present on the same day or the following day for all regions except the Southern Great Plains, where ARs precede humid heat by about a day (Figure 1, inset plots).

Separating strong ARs from weak-to-moderate ones shows an enhancement of AR/humid-heat interaction probability with increasing AR intensity for the Southwest, Midwest, and Northeast (Figure 2a). Conversely, the absence of an AR translates to lower-than-normal humid-heat risk in the Northern Great Plains, Midwest, and Northeast, while a risk reduction is also seen for the case of strong ARs in the Southeast. We then test the meaningfulness of the AR/humid-heat interaction more rigorously by comparing AR extent on regional peak humid-heat days to that on a set of days with identical 500-mb geopotential-height [Z500] anomalies — in other words, we control for the possibility that strong ARs simply occur in tandem with amplified ridges. With this effect accounted for, more-extensive coverage of ARs over a region is still found to be associated with a higher probability of humid heat for the same three northern regions that stand out by other measures: the Northern Great Plains, Midwest, and Northeast (Figure 2b). ARs that extend over 50% or more of these regions are at least 2 times as likely to occur in close proximity to humid heat, for the same Z500 anomaly, versus no- or small-AR situations. Spatially extensive ARs are rare in the Northwest and Southwest, but there correlate negatively with humid-heat occurrence.

To better visualise the meteorology leading to the summary statistics presented above, we map AR and Tw composites for regional peak humid-heat days, thus aiming to illuminate the centroids of AR/humid-heat interaction for each region. Coherent large areas with high AR probabilities are again seen in association with humid heat, especially across the entire Great Plains, Midwest, and Northeast (Figure 3). The latter two also have highly spatially correlated humid heat, with most of each region exceeding the Tw 90th percentile simultaneously. Maximum anomalies of humid heat are generally located several hundred km from the AR center points, except in the Southeast and Southwest where the two are nearly co-located.

Lastly, because ARs typically involve positive anomalies of both precipitation and IVT, it is natural to ask whether the interactions we describe can be satisfactorily explained by either of the latter variables alone. Repeating the humid-heat risk-ratio analysis for precipitation and IVT separately (Figure 4) indicates that where interaction probabilities are largest (and especially in the northern tier of the US from Montana to Maine), ARs have an additional explanatory power for humid-heat risk; in other words, the relative risk of AR/humid-heat interaction is significantly greater than for either precipitation/humid heat or IVT/humid heat interactions. Also notable in Fig. 4a is the important humid-heat role played by precipitation from storms in the arid Southwest (Speizer et al. 2022), much of which is connected to the slow broad (i.e. non-AR) intrusion of moisture and related enhanced convection of the North American Monsoon (Adams & Comrie 1997).






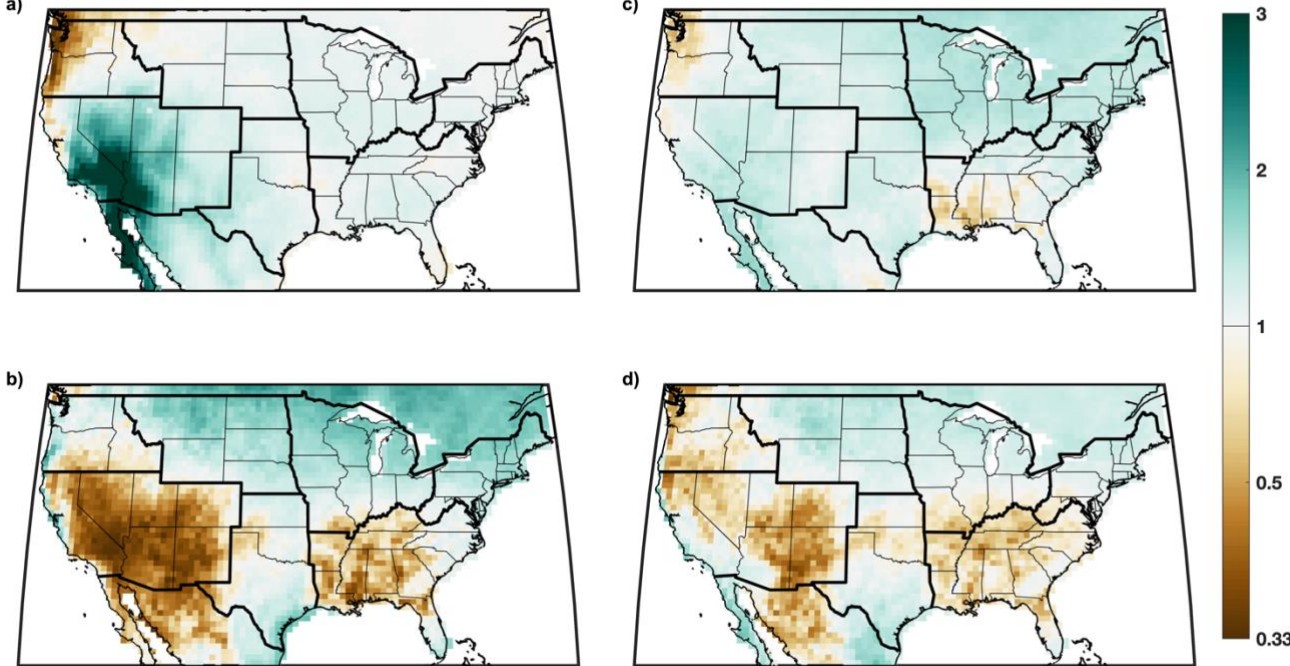

*Figure 4: Relative risk of humid heat conditioned on precipitation and IVT*

*a) Relative risk of >1 mm daily precipitation occurring within 1 day and 100 km of a humid-heat day at each gridcell. (b) Ratio of AR/humid heat relative risk (Figure 1) to precipitation/humid heat relative risk. (c,d) Same as (a,b) but for IVT.*

## 3.2 Multivariate timeseries for the Midwest

Motivated by the intra-regional coherence and high probability of AR/humid-heat interaction in the Midwest, we now focus more closely on the timeline and variables involved there (Figure 5), with analogous figures for other regions in the Appendix (Figures A1-A6). First, expanding upon Fig. 3, we examine composites one day before and after regional peak humid-heat days. As components of humid heat, we plot daily maximum temperature and specific humidity, and as AR signatures, we plot daily mean precipitation and total IVT (Figure 5). We observe here the simultaneous development of the AR and Tw anomalies as they shift eastward, with the Tw maximum anomaly always slightly ahead of the AR, echoing Fig. 1. A coherent AR structure extends into the Midwest from Texas, suggesting long-range (>1000 km) vapour transport, indicated also by Fig. 5c,f,i and agreeing with previous AR case studies in this vicinity (Gimeno et al. 2021; Lavers &





Villarini 2013). The greatest relative risk of precipitation is observed several hundred km from the maximum humid heat
anomaly, in a poleward direction perpendicular to the AR axis.

All of the above relationships are finally distilled, in a regional-average sense, into timeseries of multiple variables
for the Midwest (Figure 6). We find that although peak values of AR probability and IVT amount are sustained for two
consecutive days, the decrease of dry-bulb temperature due to the shifting position of the ridge-trough system causes
maximum Tw to occur on the first day of the pair. A positive anomaly of net surface longwave radiation on the peak Tw day
more than compensates for a decrease in net surface shortwave radiation, presumably due to cloudiness and/or water-vapour
feedbacks associated with the growing humidity and precipitation likelihood.

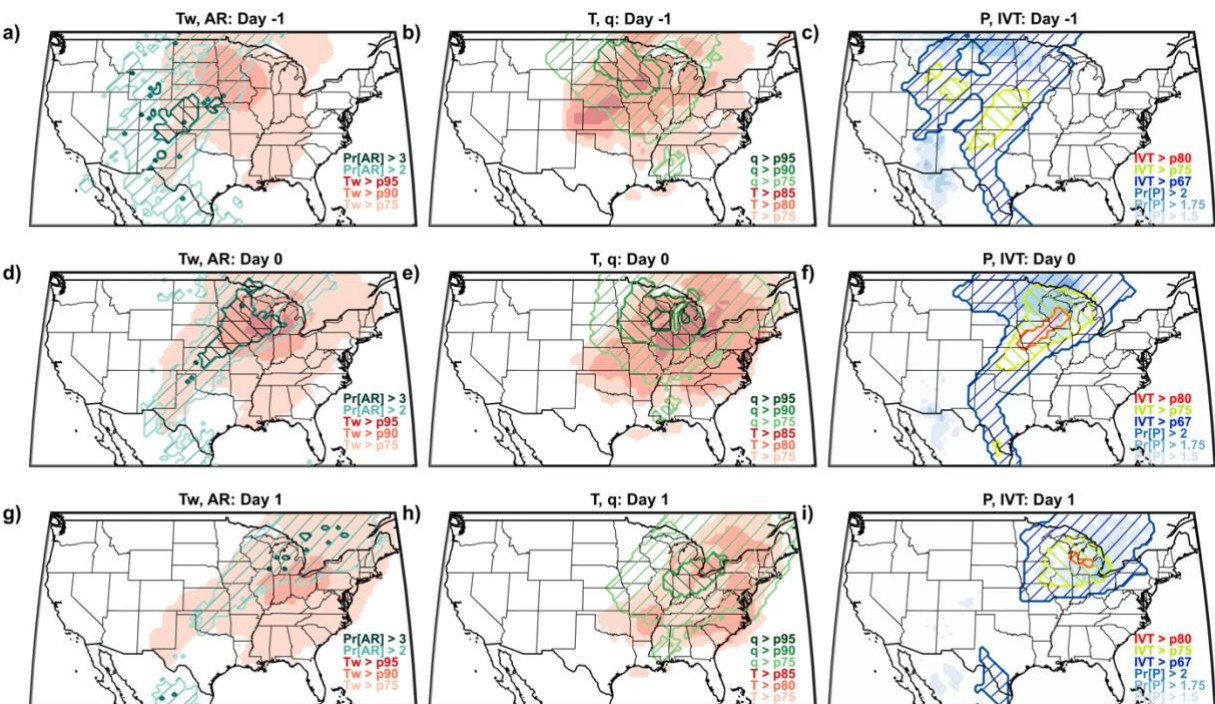

*Figure 5: Spatiotemporal progression of Midwest AR- and humid-heat-related quantities*

*(a,d,g) Tw percentiles (shaded) and AR probabilities (hatched) for the Midwest for 1 day prior to a peak humid-heat day,
the peak day, and 1 day afterward. Shading shows where composited mean Tw exceeds the May-September 95th percentile
(dark red), 90th percentile (red), or 75th percentile (light red). Contours indicate where the relative likelihood of a nearby AR
for these composited events exceeds 3 (dark teal) or 2 (light teal). (b,e,h) As in (a,d,g) but for temperature [T] and specific
humidity [q], each with intervals representing the 95th, 90th, and 75th percentiles. (c,f,i) As in (a,d,g) but for precipitation [P]
and integrated vapour transport [IVT], with intervals for the former representing the 75th and 67th percentiles, and the latter
the 90th, 75th, and 67th percentiles. Gridcells with mean May-September precipitation likelihood <10% (primarily in
California) are masked for reliability.*





## 4 Discussion and conclusions


In much of the US, we find that warm-season ARs are likely to be associated with preceding humid heat, and more specifically with a heat-then-flood timeline — a relationship that derives from the typical orientations and trajectories of mid-latitude synoptic weather systems, and that aligns with earlier results for several temperate climate zones including the Midwest (Zhang & Villarini 2020; Sauter et al. 2023). Our analysis also suggests that the AR/humid-heat connection is due

principally to ARs' water-vapour transport, at least east of the Rocky Mountains, where spatially widespread Tw extremes are likeliest to occur simultaneously with high IVT but moderate precipitation (Figure A7).

Focusing on the Midwest, broader hemispheric context reveals that southerly low-level flow over the region — which has a previously demonstrated humid-heat importance (Raymond et al. 2017) — is attributable to quasi-stationary planetary waves of wavenumber 5, which increase both temperature and moisture through a combination of advective and

radiative processes (Lin & Yuan 2022). Our work ties this mechanistic view to the detailed regional statistics of Zhang and Villarini (2020) by showing that southerly low-level flow is frequently manifest as ARs, and that these ARs mostly occur on the west flank of a ridge, resulting in precipitation that tends to lag humid heat (Figure 5). Intense IVT and precipitation on the west flank may even contribute to ridge amplification via ascent and condensational warming (Pfahl et al. 2015), and indeed this was found to be an important factor in the 2021 Western North America heat wave by several recent papers (Mo

et al. 2022; Loikith & Kalashnikov 2023). In that case, an AR landfalling in southern Alaska transported anomalous heat and moisture to the nascent ridge over British Columbia, amplifying it through both a sensible-heat effect and a water-vapour radiative feedback effect. Despite the exceptional anomalies involved, this meteorological picture, including the geographic offset between the landfall location and the peak temperature anomaly, reflects the general process separation between ARs and humid heat in the Northwest (Figure 1) and provides a glimpse of its physical basis. The separation represents a valuable

joint-risk reduction for the Northwest, as also in the Southeast and Southwest, which we suggest have distinct origins: in the former, it may be linked to the dynamics of the summertime westward expansion of the Bermuda High (Luo et al. 2021), and in the latter to the diffuse and sporadic nature of Southwest US warm-season moisture incursions generally not meeting the Guan-Waliser AR definition used here (Slinskey et al. 2020; Guan & Waliser 2019; Adams & Comrie 1997). A more in-depth study could consider such subregional variations, of the sort apparent from Fig. 1, in more detail.








**Figure 6: Midwest multivariate timeseries**

*For Midwest peak humid-heat days, composited daily timeseries of (a) AR probability and of May-September percentiles of (a) wet-bulb temperature; (b) temperature and specific humidity; (c) total integrated vapour transport and precipitation; (d) 0-5 cm soil moisture and evaporation; (e) surface net downward shortwave and net longwave radiation.*

285        While warm-season ARs are relatively common across much of the Midwestern and Eastern US, their contribution to extreme precipitation is mostly lower than that of cold-season ARs when assessed as a fraction of the seasonal total (Slinskey et al. 2020; Nayak & Villarini 2017). Nonetheless, they have been tied to major flood events in the Midwest, including in 2008 and 1993, the latter of which caused $31 billion (2022 USD) in damages (Budikova et al. 2010; Lavers &



Villarini 2013). Many sites in the Midwest had half or more of their 1980-2011 annual-maxima flood events associated with
ARs (Lavers & Villarini 2013). An important area for future work will be interrogating this AR-mediated humid
heat/precipitation connection more directly, including at the subdaily timescale. However, uncertainties related to the hourly
ordering of humid heat and precipitation are embedded in Figs. 5 and 6 and present a key challenge for high-temporal-
resolution precipitation analysis in this context: MERRA-2 suggests that in a composite sense precipitation and maximum
humid heat precisely coincide, while in station data precipitation is most likely to occur 6-12 hours after the humid-heat peak
(Figure A8). MERRA-2 hydrological variables, including observation-corrected precipitation, in fact fare well in
comparisons against other gridded products (Reichle et al. 2017). Relative to station observations, MERRA-2 also has the
advantage of self-consistently representing how humid heat and precipitation line up against each other and evolve in space
and time, particularly with respect to related quantities such as water vapour and its transport.

Our results emphasise distinct regional patterns across the US in the nature and strength of AR/humid-heat
interactions. In much of the country, and most notably in the northern tier, humid heat is closely linked to warm-season ARs
in a spatiotemporally coherent, process-based way. Additionally, this linkage cannot be fully explained by either IVT or
precipitation, two of ARs' signature features. Alternatively stated, in these regions, ARs integrate high IVT and a positioning
on the trailing side of high-pressure systems to contribute to increasing humid heat in the hours-to-days before the
temperature fall of an arriving trough (frequently accompanied by convective precipitation) (Kunkel et al. 2012). This
integration of effects also helps explain why the interaction signal tends to be stronger for stronger ARs, even when
controlling for ridge amplitude. We thus argue that consideration of AR dynamics can provide a valuable perspective for
future humid-heat and multi-hazard studies in this and other mid-latitude regions, particularly those studies aiming to
validate models, diagnose processes, or improve humid-heat predictions at weather-system timescales.

## Appendix A

*Figure A1: As in Figure 5 but for the Northwest region.*

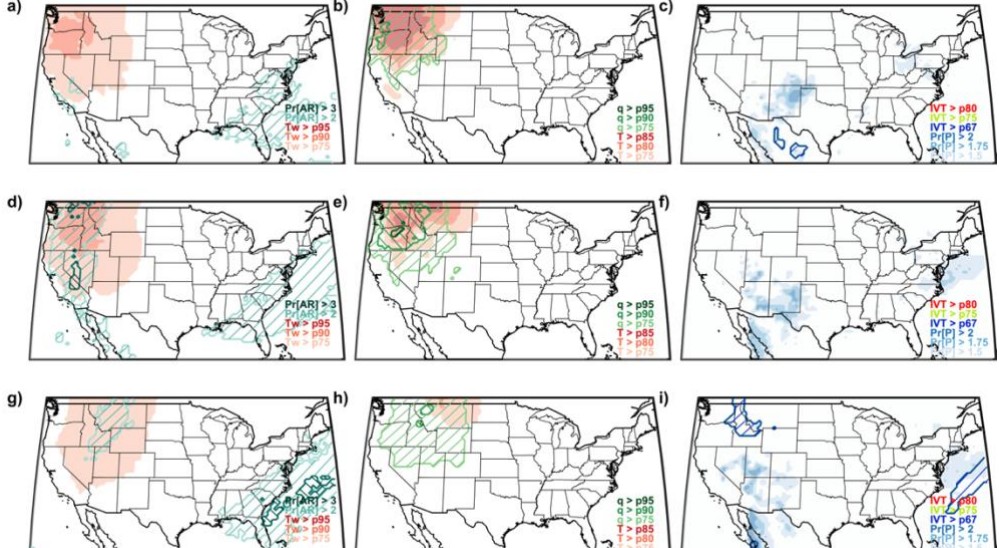




*Figure A2:* As in Figure 5 but for the Southwest region.

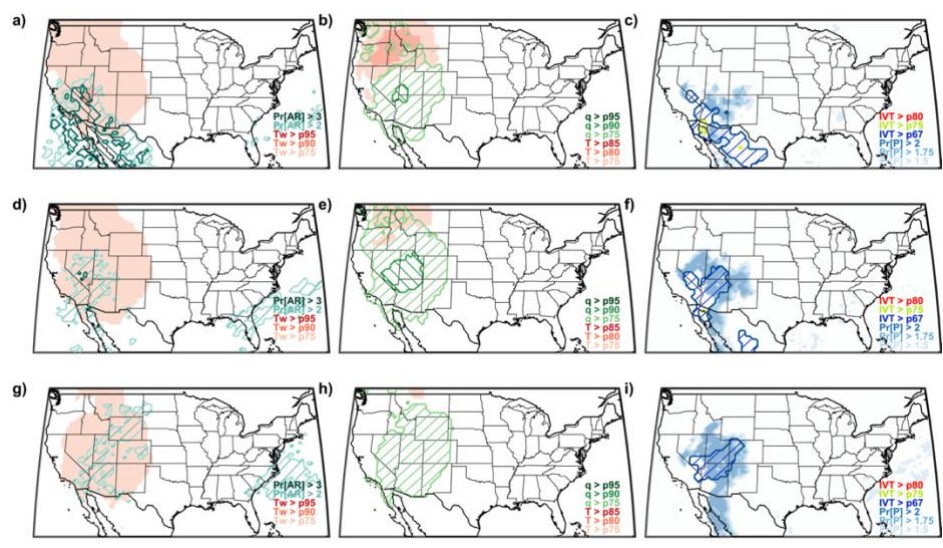

*Figure A3:* As in Figure 5 but for the Northern Great Plains region.


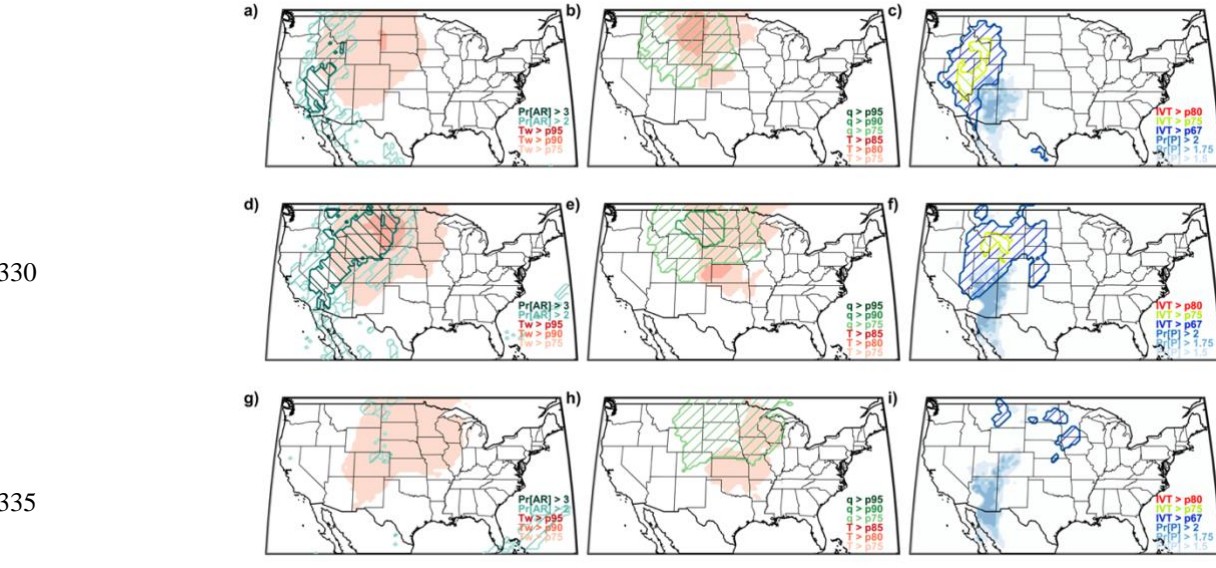





**Figure A4:** *As in Figure 5 but for the Southern Great Plains region.*

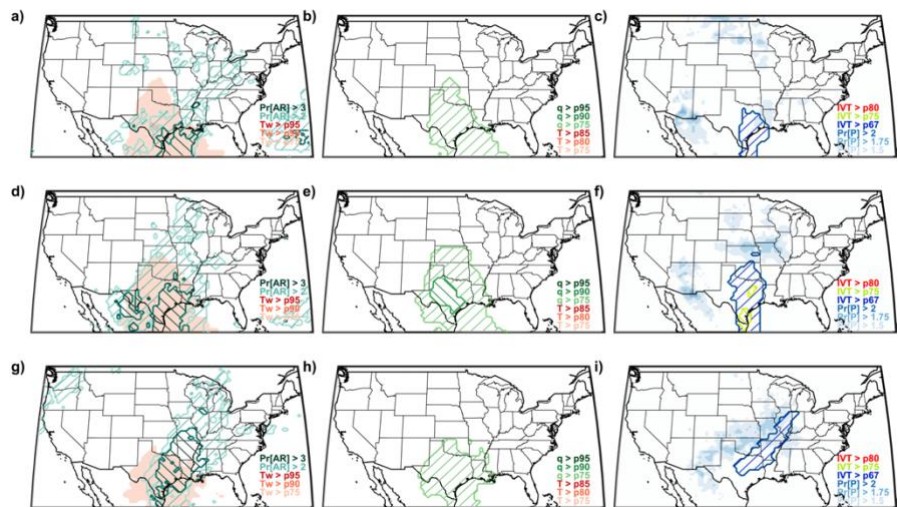


**Figure A5:** *As in Figure 5 but for the Southeast region.*

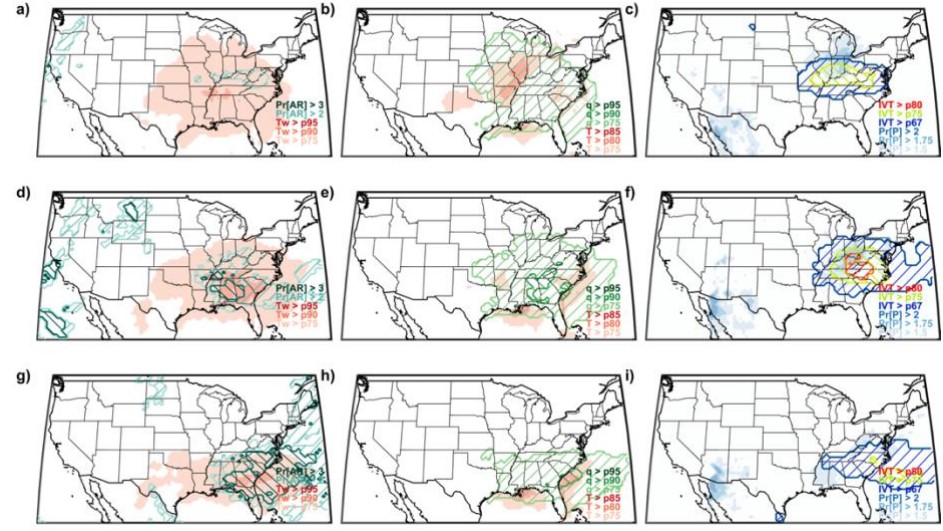






***Figure A6:*** *As in Figure 5 but for the Northeast region.*

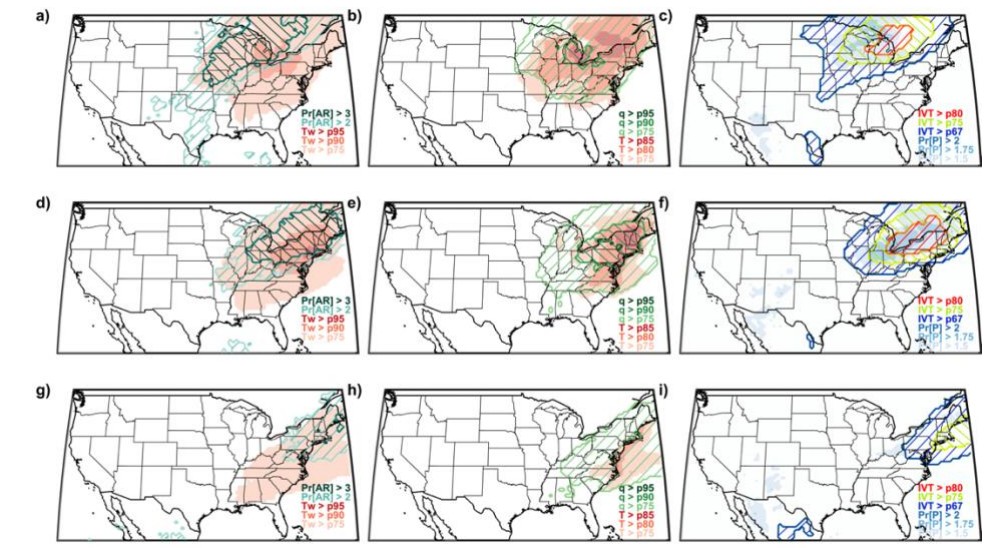

***Figure A7:*** *Mean relative extent of humid-heat gridcells in a region, partitioned according to the co-occurring daily terciles of IVT and precipitation in a 100-km radius around each gridcell in a 3-day period. The darkest shading highlights the combination of IVT and precipitation terciles most likely to co-occur with regional humid-heat days.*

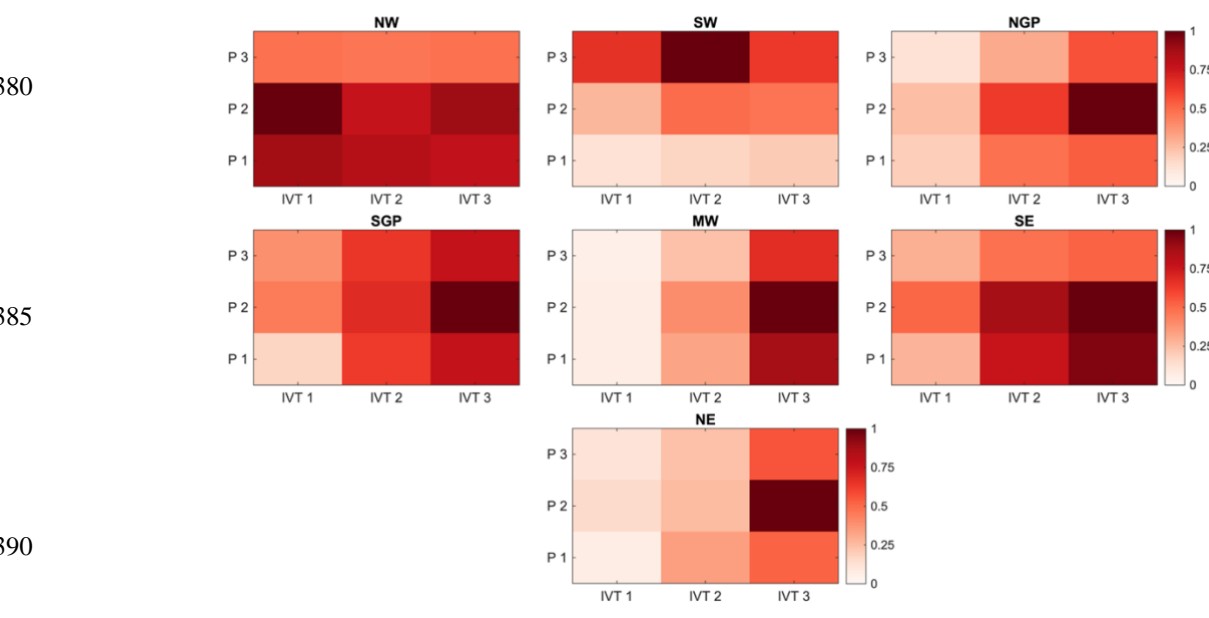




*Figure A8:* *Comparison of Tw and precipitation composite percentiles, averaged across 11 HadISD (https://www.metoffice.gov.uk/hadobs/hadisd/) weather stations and their closest MERRA-2 gridcells in the Midwest region.*
*Stations are Chicago, IL; Columbia, MO; Grand Rapids, MI; Green Bay, WI; Indianapolis, IN; Madison, WI; Minneapolis, MN; Peoria, IL; Saginaw, MI; Sioux City, IA; and Youngstown, OH.*


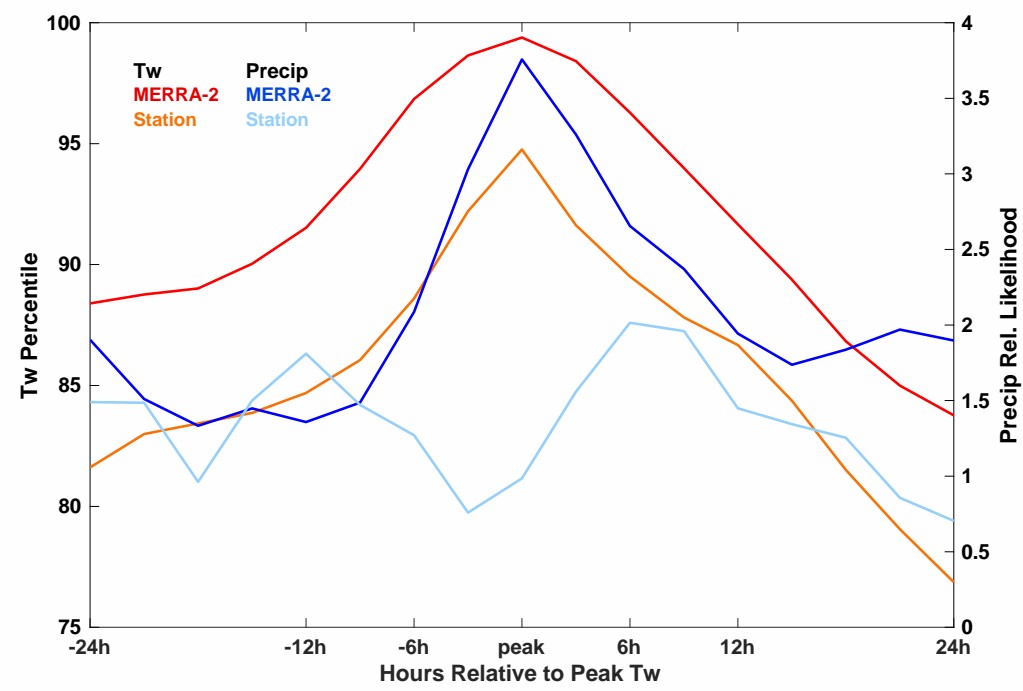

## Data availability


MERRA-2 data can be obtained from the NASA Global Modeling and Assimilation Office [GMAO]: https://disc.gsfc.nasa.gov/datasets?project=MERRA-2 (GMAO, 2015). Self-describing code for detecting ARs using the Guan-Waliser algorithm is available at https://dataverse.ucla.edu/dataverse/ar (Guan, 2021).

## Author contributions


CR initiated the study, performed the data analysis, and wrote the manuscript. AS, ES, and DW revised the manuscript. DW also contributed to providing the supporting funding.



## Competing interests

The authors declare that they have no conflicts of interest.

## Acknowledgements

A portion of the work was carried out at the Jet Propulsion Laboratory, California Institute of Technology, under a contract with the National Aeronautics and Space Administration (80NM0018D0004).

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
