# Peer review of "Linkages between atmospheric rivers and humid heat across the United States"

_EGUsphere, 2023_

## Referee Comment (RC1)

**Linkages between atmospheric rivers and humid heat across the United States**

**Authors:** Colin Raymond, Anamika Shreevastava, Emily Slinskey, and Duane Waliser

**1. Summary**

In this paper, Raymond et al, 2023 disentangle the relationship between Atmospheric Rivers (ARs) and humid heat occurrence over different regions in the US during the warm season (May to September). This approach considers peaks of wet-bulb temperature and computes the probability of occurring when detecting an AR at grid point and region level. This is also done for two other variables that typically represent ARs: precipitation and Integrated Water Vapour (IVT). Moreover, composites before and after the peak are calculated for key quantities related to humid-heat and ARs, allowing us to infer from the statistical relation which processes are key for these events.

**2. General comment**

I find this paper very interesting. It was quite easy to understand and enjoyable to read. The proposed method to assess such interaction between humid-heat and ARs focuses on the peak of humid-heat in order to examine the processes that cause humid-heat extremes rather than their maintenance. My main issue was with the "extent of the Data and methods". I was expecting a little bit more of explanation (e.g. selection of humid-heat days methodology, more detailed explanation of Relative Risk metric). There are some aspects that the authors should address before this paper can be published in NHESS. I will list them here, together with some minor/technical corrections.

**3. Specific comment**

**L10-11:** Consider rewriting the second sentence of the abstract to improve readability.

**Section 2.2 (L64-69):** What is the domain used for the AR-detection algorithm? The algorithm has geometric criteria, does it have a minimum AR extension threshold? If the domain where the detection algorithm is applied only considers the continental US, this domain can miss a significant amount of ARs due to its geometric criteria. Moreover, it can affect the results, especially in the Western US or areas close to the boundaries. If the AR detection domain used does not take this into account, consider applying the AR detection algorithm in a larger domain where the geometric criteria will not prevent detecting all the ARs. What one could do is to check if the number of detected ARs is similar to the AR Catalogue by Guan [1].

In **Section 2.3** it is explained how peak humid-heat days are selected. This method seems very restrictive, but is justified to explore the processes that lead to cause the humid-heat rather than maintaining it. Despite that, it would be good to know if the results are sensitive to the thresholds used for the "peak" framing or without the "peak" framing at all; consider including the sensitivity tests for this methodology if you have them.

**L94-95**: the Tw percentiles are computed over a 30-day-smoothed climatology. Why is this smoothing necessary to compute the $95^{th}$ percentile? Then, to define if a day is above the $95^{th}$ percentile from the smoothed climatology, you use the daily values without smoothing, which percentage of the total days fall above this threshold? Moreover, a 30-day smoothing seems

too strong, can you justify why 30 days and not 7 or 15 days, for example?

**L121-122:** What do the colours in brackets mean in the section a) of the caption for Figure 2? I thought the colours represented different US regions, it is confusing, consider removing them if they do not provide any useful information.

**L122-123:** The caption for Figure 2 section b) is not clear. Seems confusing when compared to what is written in section 2.4 (L132-135). Consider writing here what is shown in the Figure, but also make sure that it is consistent with section 2.4 (L132-135).

**L131-132**: Relative Risk metric is introduced and widely used in the paper, consider going more in-depth with the explanation of how it is calculated. And if possible, add a reference as well, this will facilitate interpretation of the results, especially for people not used to it. I would mention that "the particular sets of days" correspond to the peak humid-heat days, as this selection of days is always used in the calculation of relative risk. You could mention here that this is not only done for AR/humid-heat, but also for the precipitation threshold (1 mm) and IVT (Figure 4). Specify, which thresholds are used for these two variables, the precipitation threshold is described in Figure 4 caption, but for IVT has not been explained.

**L149/L223:** Relative likelihood is relative risk? Consider using one terminology or introduce this term in the methods section 2.4 (L131).

**L149:** Why the AR relative likelihood is within 2 days? In methods, Section 2.4 (L129) is stated to be within 1 day.

**L167:** change 500-mb to 500 hPa, as it is in the Methods.

**L172:** The results in the western regions can be sensitive to the AR detecting algorithm as mentioned in my comment for Section 2.2. Please, make sure your results are not limited by this issue.

**L194:** Which is the threshold of IVT used for the IVT/humid-heat relative risk? Consider writing this in the caption of the figure, but also in the Methods section.

**L200:** What is total IVT? Could you explain how it is computed? I have seen total IVT as the integration of an IVT vertical section across an AR to calculate the total amount of moisture an AR transports. I assume here this is not the case, please explain what it stands for.

**L207-209:** Here you state: "the decrease of dry-bulb temperature due to the shifting position of the ridge-trough system causes maximum Tw to occur on the first day of the pair", the Tw occur the first day of the pair, because the methodology on selecting peak Tw days forces to be this way. You could say that the day after the Tw peak (or the second day of the pair) the dry-bulb temperature decreases due to the shifting position of the ridge-trough, but the specific humidity (q) remains as high as the first day of the pair (or Tw peak day). I think you cannot imply causality in this case.

**L221:** Here, do the AR probabilities stand for the relative risk? I would refer to relative risk when corresponding, consider using the same therminology used in the Methods Section 2.4 to avoid confusion.

**L221&223:** First is used "hatched" to make reference to the tale colour information in subplots a,d,g and later is described as "contours". I believe it refers to the same thing, I would use the same wording to avoid confusion.

**4. Technical correction**

When referencing figures, the authors used 2 different forms, abbreviation as "Fig. ##" when added directly to the text or without abbreviation like "Figure ##" when added in brackets, I would be consistent and only use one form in the manuscript.

**Figures 1 and 3** do not present any text in the colorbar (like "Relative risk of AR/humid-heat interaction"). I would be clear in the plots what the colour values stand for, not only in the caption of the Figures.

**Figure 2:** Both y-axis for subfigures a) and b) have the same label, but stand for two different relative risks (the second is controlled by z500) as described in the caption. Consider using different y-axis labels for clarity.

**Figures 3 and 5:** In the green labels are written "Pr[AR]" and in Figure 5 the blue labels are written "Pr[P]". I understand they stand for relative risk, but I would rather rewrite them as RR[AR] and RR[P]. Pr[] can be confusing as it is used in other places with a different meaning.

**Figure 4:** Consider putting labels on top of each subplot for clarity, as it is done for Figure 5.

**L221&223 (Figure 5 caption):** The caption of subplots (b,e,h) and (c,f,i) are a bit confusing. The information in the caption does not always correspond to what is written in the legend at each subplot. Also, you write "as in (a,d,g)", but instead of showing relative likelihood, you show percentiles. I would recommend making this caption description clearer not to confuse the reader.

**5. References:**

**[1]** GUAN, BIN, 2022, "[Data] Global Atmospheric Rivers Database, Version3", https://doi.org/10.25346/S6/YO15ON, UCLA Dataverse, V3

---

## Author Comment (AC1)

**Reviewer 1**

*In this paper, Raymond et al, 2023 disentangle the relationship between Atmospheric Rivers (ARs) and humid heat occurrence over different regions in the US during the warm season (May to September). This approach considers peaks of wet-bulb temperature and computes the probability of occurring when detecting an AR at grid point and region level. This is also done for two other variables that typically represent ARs: precipitation and Integrated Water Vapour (IVT). Moreover, composites before and after the peak are calculated for key quantities related to humid-heat and ARs, allowing us to infer from the statistical relation which processes are key for these events.*

*I find this paper very interesting. It was quite easy to understand and enjoyable to read. The proposed method to assess such interaction between humid-heat and ARs focuses on the peak of humid-heat in order to examine the processes that cause humid-heat extremes rather than their maintenance. My main issue was with the "extent of the Data and methods". I was expecting a little bit more of explanation (e.g. selection of humid-heat days methodology, more detailed explanation of Relative Risk metric). There are some aspects that the authors should address before this paper can be published in NHESS. I will list them here, together with some minor/technical corrections.*

**We would like to thank the Reviewer for their efforts in summarizing our study and recommending improvements to it. We have included more detailed description of the methods, as well as made various other changes. Please find below further details on what we have changed and added (text in red). Quoted reviewer commenters are in *italics,* and our responses are in bold.**

*L10-11: Consider rewriting the second sentence of the abstract to improve readability.*

**We have trimmed several words to make the sentence simpler:**
"Process-linked connections between these two extremes, particularly those which cause them to occur close together in space or time, are of special concern for impacts."

*Section 2.2 (L64-69): What is the domain used for the AR-detection algorithm? The algorithm has geometric criteria, does it have a minimum AR extension threshold? If the domain where the detection algorithm is applied only considers the continental US, this domain can miss a significant amount of ARs due to its geometric criteria. Moreover, it can affect the results, especially in the Western US or areas close to the boundaries. If the AR detection domain used does not take this into account, consider applying the AR detection algorithm in a larger domain where the geometric criteria will not prevent detecting all the ARs. What one could do is to check if the number of detected ARs is similar to the AR Catalogue by Guan [1].*

**The AR criteria include an extent of 2000 km in length, and a length/width ratio >2, in addition to several IVT requirements as detailed in Guan & Waliser 2019. Our language was perhaps unclear; we simply look through the Guan-Waliser AR catalogue for AR**

presence/absence at each gridcell for each timestep. The catalogue was produced at a global scale, so for example an AR day at a gridcell in northern Washington State can be associated with an AR that is primarily located outside our analysis domain, such as over the ocean or British Columbia. In other words, the US domain that we use does not restrict the ARs included — all portions of ARs that affect US points are included in our analysis.

**We have modified the text to make this catalogue usage clearer:**
"Using the Guan-Waliser AR catalogue, we subsequently define AR gridcell-days as those for which an AR is present at a gridcell for at least two of that day's four timesteps. The entire AR need not fall within the US domain, as the catalogue is defined globally and we evaluate AR occurrence gridcell-by-gridcell."

*In **Section 2.3** it is explained how peak humid-heat days are selected. This method seems very restrictive, but is justified to explore the processes that lead to cause the humid-heat rather than maintaining it. Despite that, it would be good to know if the results are sensitive to the thresholds used for the "peak" framing or without the "peak" framing at all; consider including the sensitivity tests for this methodology if you have them.*

**This is a good idea. The new Figure A12 (see below) presents the same analysis as in Figure 5 but using all humid-heat days (i.e. all days with wet-bulb temperature exceeding the 95th percentile). Comparing the two figures reveals that the peak definition has little impact on the core results or their interpretation. We retain the original Figure 5, however, to ensure that we focus on processes leading to the build-up of humid heat rather than sustaining it.**

**Also of note, we also have now included a figure (Figure A1) illustrating the "peak days" method, in response to a suggestion from Reviewer 2.**

[Figure]

**Figure A1: Illustration of the definition of a peak heat-stress day. Days marked 1 and 2 satisfy the requirements of having the highest Tw value within 3 days on either side, as well as Tw having been below the 90th percentile within the preceding 3 days, while day 3 does not. As stated in the text, these requirements can apply to data from an individual gridcell or to a regional (spatial) mean.**

**Figure A12: As in Figure 5 but for all humid-heat days.**

[Figure]

*L94-95: the Tw percentiles are computed over a 30-day-smoothed climatology. Why is this smoothing necessary to compute the 95$^{th}$ percentile? Then, to define if a day is above the 95$^{th}$ percentile from the smoothed climatology, you use the daily values without smoothing, which percentage of the total days fall above this threshold? Moreover, a 30-day smoothing seems too strong, can you justify why 30 days and not 7 or 15 days, for example?*

**Our text was unclear, and actually the smoothing (which we had computed at an earlier stage of the analysis) does not factor into our final percentile calculation. The revised text has been simplified and corrected to read as follows:**
"We compute Tw percentiles for each day at each gridcell against the climatology of the surrounding 30 days, then define a 'humid-heat day' as a day with Tw above the 95$^{th}$ percentile."

*L121-122: What do the colours in brackets mean in the section a) of the caption for Figure 2? I thought the colours represented different US regions, it is confusing, consider removing them if they do not provide any useful information.*

**Yes, the colors in the figure represent different regions; the colors in parentheses in the caption erroneously referred to a previous version of the figure, and have been removed.**

***L122-123****: The caption for Figure 2 section b) is not clear. Seems confusing when compared to what is written in section 2.4 (L132-135). Consider writing here what is shown in the Figure, but also make sure that it is consistent with section 2.4 (L132-135).*

**Thank you for noting this. Upon review, we realized that the caption title was also slightly confusing and unrepresentative. The corrected caption reads:**
"Figure 2: Relative risk of humid heat by AR intensity and extent
a) For each region, the relative risk of a humid-heat day that has no AR within 1 day and 100 km ("nearby"); with an AR of category 1-3 nearby; and with an AR of category 4-5 nearby. b) Relative risk of humid heat, normalised by regional Z500 anomalies (see Methods), for different AR extents. Note that most regions lack any days with >80% regional AR coverage."

**We also revisited the section 2.4 text and revised it as follows:**
"As an additional metric for assessing how ARs and humid heat are connected, we compare two sets of days: one comprising all regional-humid-heat days, the other comprising a random selection of non-regional-humid-heat warm-season days with identical regional-mean 500-hPa geopotential height [Z500] anomalies. In other words, normalised by Z500 anomalies, we ask whether days with larger AR extents are more likely to experience humid heat within one day before or after."

***L131-132****: Relative Risk metric is introduced and widely used in the paper, consider going more in-depth with the explanation of how it is calculated. And if possible, add a reference as well, this will facilitate interpretation of the results, especially for people not used to it. I would mention that "the particular sets of days" correspond to the peak humid-heat days, as this selection of days is always used in the calculation of relative risk. You could mention here that this is not only done for AR/humid-heat, but also for the precipitation threshold (1 mm) and IVT (Figure 4). Specify, which thresholds are used for these two variables, the precipitation threshold is described in Figure 4 caption, but for IVT has not been explained.*

**We agree that the previous text was somewhat vague and have revised and expanded it:**
"Relative risk in general refers to the risk of an event of interest in a certain case relative to its risk in a control case; here, it refers to the computed probability of ARs near peak humid-heat days (i.e., of AR/humid-heat interaction) versus the probability which would be expected if ARs and humid heat were randomly distributed relative to one another throughout the warm season."

**We have added a sentence mentioning that we do the analogous computation for precipitation and IVT, including the thresholds used:**
"We analogously compute relative risk for precipitation/humid heat and IVT/humid heat, using the thresholds of 1 mm/day for precipitation and the local 75th percentile for IVT."

**This IVT threshold is now mentioned in the Figure 4 caption as well.**

***L149/L223****: Relative likelihood is relative risk? Consider using one terminology or introduce this term in the methods section 2.4 (L131).*

**We have now standardised the wording by changing all instances of 'relative likelihood' to 'relative risk' and of 'likelihood' to 'probability', as we agree that using multiple and related terms for these concepts is unnecessarily confusing.**

*L149: Why the AR relative likelihood is within 2 days? In methods, Section 2.4 (L129) is stated to be within 1 day.*

**Thanks -- this was another outdated caption, and has now been fixed.**

*L167: change 500-mb to 500 hPa, as it is in the Methods.*

**Corrected.**

*L172: The results in the western regions can be sensitive to the AR detecting algorithm as mentioned in my comment for Section 2.2. Please, make sure your results are not limited by this issue.*

**We agree that this is an important issue, but due to our usage of the globally defined Guan and Waliser catalogue (as described above), our analysis is not geographically limited. We have double-checked and confirmed that there are no artificial constraints of this sort introduced in our code — for each gridcell, all ARs that affect it are included.**

*L194: Which is the threshold of IVT used for the IVT/humid-heat relative risk? Consider writing this in the caption of the figure, but also in the Methods section.*

**As stated above, we have added to section 2.4 that we use the 75th percentile of IVT for the relative-risk calculation, and we also state this in the Figure 4 caption.**

*L200: What is total IVT? Could you explain how it is computed? I have seen total IVT as the integration of an IVT vertical section across an AR to calculate the total amount of moisture an AR transports. I assume here this is not the case, please explain what it stands for.*

**'Total IVT' is a term taken from the Guan-Waliser AR catalogue and refers to the sum of the (vertically integrated) north-south and east-west components. Because this is more of a technical detail, we have dropped the adjective 'total' for clarity, and simply call it 'IVT'.**

*L207-209: Here you state: "the decrease of dry-bulb temperature due to the shifting position of the ridge-trough system causes maximum Tw to occur on the first day of the pair", the Tw occur the first day of the pair, because the methodology on selecting peak Tw days forces to be this*

*way. You could say that the day after the Tw peak (or the second day of the pair) the dry- bulb temperature decreases due to the shifting position of the ridge-trough, but the specific humidity (q) remains as high as the first day of the pair (or Tw peak day). I think you cannot imply causality in this case.*

**Indeed, our phrasing did not properly account for the complete methodology, and especially the fact that Tw peaks on the central day of Figure 6 by definition. We took the Reviewer's suggestion in the revised text:**
"We find that although peak values of AR probability and IVT amount are sustained for two consecutive days, dry-bulb temperature decreases on the second day of the pair due to the shifting position of the ridge-trough system, while specific humidity remains nearly as high as on the first day."

*L221: Here, do the AR probabilities stand for the relative risk? I would refer to relative risk when corresponding, consider using the same therminology used in the Methods Section 2.4 to avoid confusion.*

**We have changed the wording throughout the manuscript to 'relative risk' to ensure there is no confusion between this and other metrics.**

*L221&223: First is used "hatched" to make reference to the tale colour information in subplots a,d,g and later is described as "contours". I believe it refers to the same thing, I would use the same wording to avoid confusion.*

**The hatching is indeed only within the contours, and so we have changed both instances to 'hatched contours' for clarity.**

*4. Technical correction*
*When referencing figures, the authors used 2 different forms, abbreviation as "Fig. ##" when added directly to the text or without abbreviation like "Figure ##" when added in brackets, I would be consistent and only use one form in the manuscript.*

**These cases have all been standardized to "Figure #".**

*Figures 1 and 3 do not present any text in the colorbar (like "Relative risk of AR/humid-heat interaction"). I would be clear in the plots what the colour values stand for, not only in the caption of the Figures.*

**We have added a colorbar label (reading "Relative Risk") in Figure 1. In Figure 3, we enlarged the colorbar label because it was quite small and easy to miss:**

[Figure]

*Figure 2: Both y-axis for subfigures a) and b) have the same label, but stand for two different relative risks (the second is controlled by z500) as described in the caption. Consider using different y-axis labels for clarity.*

**We have changed the y-axis label of panel (b) of Figure 2 to "Z500-Normalized Relative Risk of Extreme Tw".**

*Figures 3 and 5: In the green labels are written "Pr[AR]" and in Figure 5 the blue labels are written "Pr[P]". I understand they stand for relative risk, but I would rather rewrite them as RR[AR] and RR[P]. Pr[] can be confusing as it is used in other places with a different meaning.*

**Thank you for the suggestion. The figure labels have been corrected in this way for Figures 3 and 5, as well as for Figures A2-A7.**

*Figure 4: Consider putting labels on top of each subplot for clarity, as it is done for Figure 5.*

**We have added subplot labels to Figure 4:**

[Figure]

*L221&223 (**Figure 5 caption**): The caption of subplots (b,e,h) and (c,f,i) are a bit confusing. The information in the caption does not always correspond to what is written in the legend at each subplot. Also, you write "as in (a,d,g)", but instead of showing relative likelihood, you show percentiles. I would recommend making this caption description clearer not to confuse the reader.*

**We appreciate the catch — the (c,f,i) subplot labels referred to a slightly outdated figure version. We have revised them and added new text to be clearer without excessively lengthening the caption. The revised portion of the Figure 5 caption is:**
"(c,f,i) As in (a,d,g) but for precipitation [P] and integrated vapour transport [IVT], with intervals for the former representing a relative risk of 2, 1.75, and 1.5 on composited humid-heat days, and for the latter the 80$^{th}$, 75$^{th}$, and 67$^{th}$ percentiles. These specific thresholds were chosen for visual clarity."

---

## Author Comment (AC2)

**Reviewer 2**

*First of all, I would like to apologize for the long delay in providing the review of the manuscript.*
*The authors investigate the linkages between atmospheric rivers and humid heat across the United States. For that, they use MERRA-2-based Guan-Waliser AR-detection algorithm and also daily maxima of 2-m wet-bulb temperature.*
*The manuscript is usually well written and the methodology is sound even if some points are not that clear. In my opinion, the manuscript can be accepted after minor revision mentioned below.*

**We appreciate the Reviewer's input, and recognize that all of us are overburdened with these sorts of tasks. We are glad for the overall positive assessment and respond to the Reviewer's specific concerns below, which have been helpful in validating our results and clarifying our thinking, thus improving the manuscript considerably (both through revising the text and through the addition of several new supplemental figures). Please find below further details on what we have changed and added (text in red). Quoted reviewer commenters are in *italics*, and our responses are in bold.**

*Even though the authors acknowledge the fact that AR have different phenomenology, I was wondering if the authors can enlarge the introduction relatively to that matter. In addition, can the authors also comment on the different between AR in the cold and warm season? And also, the different types of ARs that can reach different areas of the US?*

**We have added several more references on different types of ARs, especially about cold versus warm season varieties and regional distinctions, and have expanded this part of the introduction accordingly. The revised text reads:**
"ARs can be further divided along dimensions including moisture versus wind-dominated (Gonzales et al. 2020), transient versus quasi-stationary (Park et al. 2023), and tropical versus extratropical (Reid et al. 2022), as well as other distinct regional characteristics — all differences which affect ARs themselves and their impacts (Park et al. 2021; Guan & Waliser 2019; Nayak & Villarini 2017). This variety of systems falling under a single broad heading is also the case for other important climate phenomena, such as droughts (Haile et al. 2019). Although the first-described and best-known AR types occur in the extratropical cold season, warm-season varieties can have a substantial imprint on regional hydroclimate (Slinskey et al. 2020). To take North America as an illustrative case, about half of summer extreme-precipitation days in the Eastern and Central US are caused by ARs. Summer ARs over the US originate from the Pacific Ocean or (especially) the Gulf of Mexico, and tend to be weaker but wetter than their cold-season counterparts due to the higher temperatures and associated background water-vapor quantities (Slinskey et al. 2020; Neiman et al. 2008)."

*Section 2.1. the authors mentioned that they used 6-hourly data from MERRA-2, however in section 2.3 they mentioned that the use of hourly data for computing the 2-m wet-bulb temperature. I am assuming that the data here also comes from MERRA-2.*

**Yes, all the data is from MERRA-2. It has hourly data available for some variables, but the AR algorithm was only processed every 6 hours (there is relatively little hour-to-hour change in ARs). We have adjusted a few words in Sections 2.1, 2.2, and 2.3 to ensure these descriptions are clear and self-consistent.**

*Fig 1. Can you please put the name of the regions inside them? In the preset version is not very readable.*

**To improve the readability of Figure 1, we have added arrows connecting the region labels (near each inset plot) to the corresponding region on the main map. We feel this is a good solution because it avoids having duplicate labels which might be confusing or cluttered:**

[Figure]

*Fig 1. Caption - Authors need to add the information regarding the relative risk. Higher values correspond to higher risk?*

**We have expanded the Figure 1 caption to include this information:**
"Relative risk > 1 corresponds to a risk larger than that expected by chance."

*Section 2.3 Can the authors expand the explanation regarding the computation of the percentiles? Did you compute the percentile after or before the 30-day smoothing? In addition,*

*can the authors include a figure in the supplementary material explaining 3 days highest Tw value? And also, the difference between "regional" and "regional peak"?*

**The percentiles were actually computed later and independent of the smoothing, so the revised text removes mention of smoothing altogether:**
"We compute Tw percentiles for each day at each gridcell against the climatology of the surrounding 30 days, then define a 'humid-heat day' as a day with Tw above the 95$^{th}$ percentile."

**The percentiles are computed from a 30-day block surrounding each day (comprised of 30 days x 41 years = 1,230 days).**

**A definition figure is a good suggestion. We have created one as Figure A1, and we direct interested readers to it in Section 2.3. It is reproduced here for convenience:**

[Figure]

Figure A1: Illustration of the definition of a peak heat-stress day. Days marked 1 and 2 satisfy the requirements of having the highest Tw value within 3 days on either side, as well as Tw having been below the 90th percentile within the preceding 3 days, while day 3 does not. As stated in the text, these requirements can apply to data from an individual gridcell or to a regional (spatial) mean.

*Section 2.4 Did the authors use the axis of the AR, or the area of the AR? If you use the area provided by the Guan-Waliser AR-detection algorithm, then I don´t understand that a grid cell should be 100km from an AR.*

**Section 2.4 refers to distance from the edge of an AR. In other words, the gridcell in question could be within an AR, or no more than 100 km from its edge. We have rewritten and expanded this description:**
"We define as 'interaction' between ARs and humid heat those cases where humid-heat days at a gridcell occur within 1 day and 100 km of an AR. Spatially, this means a gridcell could be included within an AR, or the edge of an AR is no more than 100 km away; temporally, it means the spatial criterion is satisfied on the day before, the day after, or the same day as a humid-heat day."

*I am just wondering if having a new sub-section with a case study would also benefit the potential readers to better understand the methodology?*

**We believe that the new Figure A1 showing the peak-day definition, as well as the various text revisions prompted by the Reviewer's helpful comments, mean that a methodology-oriented example figure would not be of great additional value.**

*Section 3.1. L160 onwards. You could add a figure on characterizing the different ARs that strike the different regions? It would help a lot in understanding the results. Are they associated with Extra-tropical cyclones? They are wind vs humidity driven?*

**Thanks to this suggestion, we have added Figures A10 and A11 (copied at the bottom of this Response document). These figures show that summertime ARs in each region tend to occur between a surface low to the northwest and a surface high to the east or southeast, as is also often the case in winter (Ralph et al. 2020). The ARs are generally colocated with areas of high precipitation likelihood and high IVT. Combined with the composite map of summertime ARs affecting each region created by Slinskey et al. 2020, their Figure 6, we see that US ARs east of the Rocky Mountains tend to track along the western and northern periphery of the North Atlantic Subtropical High [NASH]. Several of these points are mentioned in the revised text:**
"Simultaneously, this flow is also often manifest as an amplified state of the warm-season Great Plains Low-Level Jet, itself often enhanced by proximity to the North Atlantic Subtropical High (Zhou et al. 2020; Budikova et al. 2010). Our work ties this mechanistic view to the detailed regional statistics of Zhang and Villarini (2020) by showing that southerly low-level flow in the Midwest is frequently classified as an AR, and that these ARs mostly occur on the west or north flank of a ridge, resulting in precipitation that tends to lag humid heat because of the usual eastward motion of mid-latitude weather systems (Figure 5)."

"In much of the US, we find that warm-season ARs are often associated with preceding humid heat, and more specifically with a heat-then-flood timeline — a relationship that derives from the typical orientations and trajectories of mid-latitude synoptic weather systems, with AR-related IVT progressing from southwest to northeast between a surface low and high (Ralph et al. 2020). Heat followed by heavy precipitation is consistent with earlier results for multiple seasons and for several temperate climate zones including the Midwest (Zhang & Villarini 2020; Sauter et al. 2023)."

**The revised Discussion also notes other regional differences in ARs and how they are likely influencing our results:**
"The tendency for ARs and humid heat to be distinct hazards in certain regions (Figure 1) can be understood through analyses of this sort. Considering first the Northwest, humid-heat days there are in fact mostly hot and dry, driven by processes (sensible heating, warm-air advection) antithetical to those associated with ARs (Raymond et al. 2017). Despite the exceptional anomalies involved, the above example, specifically the geographic offset between landfall location and peak temperature anomaly, may be illustrative in this regard. A valuable reduction of joint risk is also apparent for the Southeast and Southwest. In the Southeast, it may be linked

to the dynamics of the summertime westward expansion of the North Atlantic Subtropical High (Luo et al. 2021), which would also explain why humid heat is most unlikely near strong ARs there; in the Southwest, this joint-risk reduction may stem from the diffuse and sporadic nature of North American Monsoon moisture incursions generally not meeting the Guan-Waliser AR definition (Slinskey et al. 2020; Guan & Waliser 2019; Adams & Comrie 1997)."

**In other words, across the US there are several distinct and well-described seasonal features that can be classified as ARs when the associated moisture transport is large and organized into sufficiently long and narrow bands. For this reason, the final paragraph of the Discussion intentionally mentions ARs' characteristic features of IVT and precipitation without specifying more closely, because the details of how these are achieved in a meteorological sense vary substantially by region.**

**Classification of ARs associated with humid heat, by for example moisture versus wind dominated, is a great idea for exploration, but in this manuscript format we consider it beyond the scope of our focus.**

*2 Why the risk decreases if you go to the higher AR categories in some regions (eg. NGP, SGP and SE?)?*

**We hypothesize that this feature is partly due to the mutually exclusive nature of processes that lead to humid heat and to strong ARs in these regions, and partly due to sample-size effects. In the Southeast, for example (original Figure A5), the maximum humid heat occurs distinctly south of the maximum IVT and precipitation likelihood. In Figure 2, and in our analysis overall, we would say that our results are most easily interpretable east of the Rocky Mountains. We added a clause to note this in introducing Figure 2:**
"Separating strong ARs from weak-to-moderate ones shows an enhancement of AR/humid-heat interaction probability with increasing AR intensity for the Southwest, Midwest, and Northeast, though with some uncertainty due to sample-size effects (Figure 2a)."

**And also in the Discussion:**
"A valuable reduction of joint risk is also apparent for the Southeast and Southwest. In the Southeast, it may be linked to the dynamics of the summertime westward expansion of the North Atlantic Subtropical High (Luo et al. 2021), which would also explain why humid heat is most unlikely near strong ARs there"

*I like figure 4. Maybe the authors can explain the physical process behind the heat conditioned on precipitation and IVT?*

**Our best understanding of Figure 4 is that precipitation, IVT, and humid heat are all well-correlated in the northern tier of the country. IVT is the more important variable, but analyzing humid heat via ARs adds more value than either one, probably due to nonlinear interactions e.g. (re-)evaporation of precipitation. The Western US feels influences from systems such as the North American Monsoon that are not necessarily well suited to the AR**

**definition that is used here (although this definition is used globally and in many studies). We agree the physical processes involved are interesting and worthy of further exploration, as we now also state in the Discussion:**

"This integration of likely nonlinear effects also helps explain why the interaction signal tends to be stronger for stronger ARs, even when controlling for ridge amplitude. However, the exact physical mechanisms involved remain uncertain and a worthy subject for exploration."

*Regarding the Midwest, is this a proper AR feature, or more related with an LLJ feature? At least in late winter some of the AR there are associated with a extra-tropical cyclone : https://blog.weather.us/atmospheric-river-to-bring-heavy-rain-and-possible-flooding-to-parts-of-the-east-coast-later-this-week/*

**We agree that the Midwest and indeed all of the US east of the Rocky Mountains experience ARs that are rather different than the classic oceanic ones which affect the US West Coast, Portugal, etc. This variety of ARs (which are all classified as such according to the Guan-Waliser algorithm) is discussed in several papers, including Slinskey et al. 2020, Gimeno et al. 2021, Ralph et al. 2020, and Reid et al. 2022 (doi: 10.1175/jcli-d-21-0606.1). We have added manuscript text that cites these papers' description of how features such as active monsoon patterns or the Great Plains Low-Level Jet can exhibit AR-like characteristics despite quite diverse driving mechanisms. Later in the discussion, we have also added a reference to Higgins et al. 1997 to direct interested readers to another authoritative source on the subject:**

"An important area for future work will be interrogating this AR-mediated humid heat/precipitation connection more directly, including at the subdaily timescale, as well as the extent to which it can be considered a direct signature of the Great Plains Low-Level Jet (Higgins et al. 1997)."

**As we are focused on summer, extratropical cyclone activity is considerably weaker and less frequent, and circulations associated with the North Atlantic Subtropical High dominate AR activity east of the Rockies (Slinskey et al. 2020, Zhou et al. 2020) — see earlier response.**

*Regarding my point 8) and considering all the information provided in the manuscript, what is missing are the composites (SLP, GPT500, other variable ??) of the AR days for each one of the regions.*

**Following this excellent suggestion, composites of AR days unconditioned on any humid-heat categorization have now been produced as Figures A10-11. Figure A10 shows AR probabilities and Z500 anomalies, while Figure A11 shows IVT and precipitation percentiles. They are relevant to several responses in the document, as stated above, so please refer to those passages for our interpretation (as much as fits in the space available).**

**The new figures are reproduced here:**

**Figure A10:** Composite of AR probabilities (contours) and Z500 anomalies (shading) for all AR days at the central gridcells of each region. Light (dark) green contours indicate AR probabilities >50% (>80%), while light red (dark red) shading indicates Z500 anomalies >25 m (>50 m) and light blue (dark blue) shading indicates Z500 anomalies <-25 m (<-50 m).

[Figure]

**Figure A11:** Composite of IVT percentiles (contours) and precipitation percentiles (shading) for all AR days at the central gridcells of each region. Orange (red) contours indicate IVT percentiles >70th (>85th), while light blue (dark blue) shading indicates precipitation percentiles >60th (>80th). These values are chosen to best highlight the regions of interest.

[Figure]

**[As a reminder, composite AR days intersected with regional peak humid-heat days have already been plotted in Figure 5 and Figures A2-7.]**

New references:

Gonzales, K. R., Swain, D. L., Barnes, E. A., and Diffenbaugh, N. S.: Moisture- versus wind-dominated flavors of atmospheric rivers, Geophys. Res. Lett., 47, e2020gl090042, doi:10.1029/2020gl090042, 2020.

Haile, G. G., Tang, Q., Li, W., Liu, X., and Zhang, X., Drought: Progress in broadening its understanding, WIREs Water, 7, e1407, doi:10.1002/wat2.1407, 2019.

Higgins, R. W., Yao, Y., Yarosh, E. S., Janowiak, J. E., and Mo, K. C.: Influence of the Great Plains Low-Level Jet on summertime precipitation and moisture transport over the Central United States. J. Clim., 10, 481-507, doi:10.1175/1520-0442(1997)010<0481:iotgpl>2.0.co;2, 1997.

Park, C., Son, S.-W., and Guan, B.: Multiscale nature of atmospheric rivers, Geophys. Res. Lett., 50, e2023gl102784, doi:10.1029/2023gl102784, 2023.

Reid, K. J., King, A. D., Lane, T. P., and Hudson, D.: Tropical, subtropical, and extratropical atmospheric rivers in the Australian region, J. Clim., 35, 2697-2708, doi:10.1175/jcli-d-21-0606.1, 2022.

Zhou, W., Leung, L. R., Song, F., and Lu, J.: Future changes in the Great Plains Low-Level Jet governed by seasonally dependent pattern changes in the North Atlantic Subtropical High, Geophys. Res. Lett., 48, e2020gl090356, doi:10.1029/2020gl090356, 2020.